# The Bittersweet Effects of COVID-19 on Mental Health: Results of an Online Survey among a Sample of the Dutch Population Five Weeks after Relaxation of Lockdown Restrictions

**DOI:** 10.3390/ijerph17239073

**Published:** 2020-12-04

**Authors:** Mandy Gijzen, Laura Shields-Zeeman, Marloes Kleinjan, Hans Kroon, Henriëtte van der Roest, Linda Bolier, Filip Smit, Derek de Beurs

**Affiliations:** 1Trimbos Institute, Netherlands Institute of Mental Health and Addiction, P.O. Box 725, 3500 AS Utrecht, The Netherlands; LShields-Zeeman@trimbos.nl (L.S.-Z.); MKleinjan@trimbos.nl (M.K.); HKroon@trimbos.nl (H.K.); HRoest@trimbos.nl (H.v.d.R.); lbolier@trimbos.nl (L.B.); fsmit@trimbos.nl (F.S.); dbeurs@trimbos.nl (D.d.B.); 2Department of Child and Adolescent Psychiatry, GGZ Oost Brabant, P.O. Box 35427, 5427 ZG Boekel, The Netherlands; 3Interdisciplinary Social Science, Utrecht University, P.O. Box 80140, 3508 TC Utrecht, The Netherlands; 4Tranzo, School of Social and Behavioural Sciences, Tilburg University, P.O. Box 90153, 5000 LE Tilburg, The Netherlands; 5Department of Clinical, Neuro and Developmental Psychology and Department of Epidemiology and Biostatistics, Amsterdam Public Health Research Institute, University Medical Centers, Location VUmc, P.O. Box 7057, 1007 MB Amsterdam, The Netherlands

**Keywords:** COVID-19, stability, mental health, lockdown, positive effects

## Abstract

Previous research shows that crises can have both negative and positive mental health effects on the population. The current study explored these effects in the context of the COVID-19 pandemic after relaxation of governmental measures. An online survey was administered among a representative sample of the Dutch population (*n* = 1519) in June 2020, ten weeks after the peak of COVID-19 had passed, and five weeks after restrictions were relaxed. Participants were asked about mental health, adverse events during COVID-19, and about any positive effects of the pandemic. Most participants (80%, *n* = 1207) reported no change in mental health since the COVID-19 pandemic. This was also the case among respondents who had experienced an adverse event. Protective factors of mental health were being male and high levels of positive mental well-being. Risk factors were emotional loneliness and the experience of adverse life events. Social loneliness was positively associated with stable mental health, stressing the importance of meaningful relationships. Note that 58% of participants reported positive effects of the pandemic, the most common of which were rest, working from home, and feeling more socially connected. In summary, 10 weeks after the start of the crisis, and 5 weeks after relaxation of the restrictions, most people remained stable during the crisis, and were even able to report positive effects.

## 1. Introduction

The outbreak of the COVID-19 pandemic has affected many countries and impacted millions of lives since its onset in December 2019. As no vaccine or treatment is currently available, countries are managing the outbreak of the pandemic byleveraging existing health system and public health capacity, as well as enforcing measures to slow the spread of the virus. These measures include quarantine and stay-at-home measures, physical distancing, and movement restrictions, both within and beyond national borders. The virus, lockdown and quarantine measures, and secondary effects of the pandemic such as economic downturn, could all adversely impact mental health [1,2,3]. Initial studies from China showed higher levels of mental health problems in the general population after the start of the crisis [4,5]. Similarly, in the Netherlands, a representative study from the Netherlands among 90,000 participants showed that, at the start of the COVID-19 pandemic, approximately one-third of Dutch people were experiencing additional psychological complaints [6]. Furthermore, a group of international experts warned about the risk of the increase of suicidal behavior due to worsened mental health due to the crisis [1].

Though numerous studies have been done to gauge the early impacts of the pandemic on mental health, it remains unclear how COVID-19 will impact mental health in the longer term. For example, a longitudinal study on the impact of the corona crisis on mental well-being from the UK [7] found that, over time, symptoms of anxiety and levels of defeat and entrapment actually decreased across three waves, and well-being increased. They did notice a small rise in suicidal thoughts over time. In the Netherlands, an ecological momentary assessment study among Dutch adolescents found that initial psychosocial problems during the early stages of COVID-19 rapidly decreased after three days [8]. A recent study from England reported that although anxiety about the corona virus had a negative impact on people, the lockdown measures also had benefits, such as more working at home, at least for some participants [9].

The aim of the current study was to investigate the mental health and well-being of adults in the Netherlands ten weeks after the start of COVID-19, and five weeks after relaxation of the COVID-19 restrictions. As this is an exploratory study of a unique situation, it was difficult to set hypotheses. One could reason that, at the moment of assessment, about one-third of the Dutch population would still report worsened mental health compared to before the pandemic, because the COVID-19 pandemic was still very much present in the Netherlands, physical distancing measures and the strong request to work from home when possible were still in place, and there are realistic fears of a deep recession. On the other hand, it might be that people adapted to the new restrictions, and were relieved by the relaxation of restrictions, resulting in a better overall mental health status. An additional aim of this study was to identify supportive factors to stay mentally well during the pandemic, and if participants were able to identify any positive aspects arising from the COVID-19 pandemic.

## 2. Materials and Methods

### 2.1. Participants and Procedure

An online survey was conducted among a sample of the Dutch population from 12 to 19 June 2020. At that time, the Netherlands had moved past the peak of the COVID-19 in terms of number of daily cases, number of people in intensive care units, and number of deaths ten weeks prior. On 11 May 2020, the Dutch government relaxed protective measures by allowing primary schools, libraries, and some small businesses to reopen. From 1 June 2020, protective measures were further relaxed by allowing secondary schools, cultural institutions (e.g., cinemas/theaters), and restaurants to reopen, and nursing homes were reopened for visitors two weeks later—provided that physical distancing of 1.5 m was respected.

Participants were recruited via an online panel maintained by a Dutch external polling and research organization (I&O Research). Participants had signed up to be part of an earlier survey panel consisting of approximately 25,000 Dutch citizens. Being part of the panel, they participated in surveys on various topics on a regular basis. The invitation for this survey was sent via email, with two follow-up reminders. The survey was completely anonymous to ensure confidentiality and reliability of data.

### 2.2. Measurements

Basic demographic variables were collected, such as sex and age. Education level was measured on three levels: (1) lower education including pre-vocational education and secondary vocational education level 1; (2) middle education including secondary vocational education level 2 and up, senior general secondary education and preuniversity education; and (3) higher education including higher professional education and university education.

#### 2.2.1. Measurements Regarding Mental Health

Mental health change during COVID-19. Participants were asked to rate their mental health before the COVID-19 pandemic and at this moment (i.e., during the COVID-19 pandemic) on a 5-point scale: (1) very good, (2) good, (3) okay, (4) bad, (5) very bad. A change score was calculated to classify whether participants had a (1) worsened mental health, (2) stable mental health or (3) improved mental health.

Loneliness. The 11-item version of the Loneliness Scale [10] was used to assess both social and emotional loneliness. All items are scored on a 3-point scale: 3 ‘yes’, 2 ‘more or less’, 1 ‘no’. The emotional loneliness subscale consisting of 6 items has good validity [10] and reliability in the current sample (Cronbach’s α = 0.80). The items in this scale represent the missing of a social bond with someone or missing emotional companionship [11]. An example item: “I miss having a really close friend”. The social loneliness subscale consisting of 5 items also has been shown to have good validity [10], and reliability in the current sample (Cronbach’s α = 0.82). The items in this scale represent lacking a network of social relationships [11]. An example item: “There is always someone I can talk to about my day-to-day problems”.

Positive Mental Well-Being. The seven-item scale of the short Warwick-Edinburgh Mental Well-Being Scale (SWEMWBS) [12] was used to assess well-being. It focuses on the positive of psychological functioning for the past four weeks in the current study, such as optimism, autonomy, agency, curiosity, clarity of thought, and positive relationships. Items are scored on a 5-point scale: 1 ‘none of the time’, 2 ‘rarely’, 3 ‘some of the time’, 4′ ‘often’, 5 ‘all of the time’. Reliability in the current sample was good (Cronbach’s α = 0.84). The time period of the questionnaire was changed from two weeks to four weeks. Although this limits the comparison with other studies using the same scale, we adjusted the time period to fully include the period since the relaxation of the restrictions.

Resilience. The Brief Resilience Scale was used to measure resilience [13], which consists of six items scored on a 5-point scale: 1 ‘strongly disagree’, 2 ‘disagree’, 3 ‘neutral’, 4 ‘agree’, 5 ‘strongly agree’. It measures how people deal with stressful/adverse events in their life and how easily they bounce back [13]. Reliability in the current sample was good (Cronbach’s α = 0.89).

Suicide ideation. Suicidal ideation in the last week was assessed by the question ‘How often have you thought about taking your life in the last week?’ (‘one day’, ‘several days’, ‘more than half the days’, ‘nearly every day’, ‘never’, ‘I would rather not answer’). Results were dichotomized into yes (any suicide ideation) and no (no suicide ideation).

#### 2.2.2. Adverse Events and Possible Positive Outcome

The survey included two single items on adverse events and positive outcomes that emerged during the COVID-19 pandemic:

Adverse events experienced during the COVID-19 pandemic. Participants were asked to indicate whether they had experienced an adverse life event during the COVID-19 pandemic, such as the death of a relative, or income or job loss. The items were asked in an open-question format where participants could indicate and provide a brief description of the adverse events.

Positive aspects of the COVID-19. Participants were asked to indicate in an open-question format whether they could name any positive outcomes emerging from the COVID-19 pandemic so far, and, if so, describe the positive outcomes.

#### 2.2.3. Additional Measures

Social media use change. Participants were asked to indicate whether their social media use had changed compared to their usage before the COVID-19 pandemic started. The change score was reflected using a 5-point scale: 1 ‘increased a lot’, 2 ‘increased’, 3 ‘remained the same’, 4 ‘decreased’ or 5 ‘decreased a lot’.

### 2.3. Data Analysis

All data were analyzed using Stata version 16 (StataCorp LLC, 4905 Lakeway Drive, College Station, TX, USA) [14]. We calculated the intra-individual response variability to identify responders who engage in insufficient effort responding (IER). IER stems from participants who are unmotivated and answer questions in a set pattern (e.g., only 5 s or 1-2-3, 1-2-3). As participants were recruited via a research panel, we did not expect to identify such response patterns. Indeed, no such participants were identified and were excluded from further analyses.

The sample distribution over gender and age was compared with population data from Statistics Netherlands, and post-stratification weights were computed. All subsequent analyses were based on the weighted sample using the first-order Taylor-series linearization method as implemented in Stata version 16 [14] to obtain robust standard errors, test statistics and 95% confidence intervals; after weighting, the sample followed exactly the same distribution as the Dutch population with regard to gender and age.

Mental health change was dichotomized for further analyses: (1) worsened mental health and (2) stable mental health. The participants with self-reported improved mental health were not further analyzed as this group was too small to perform reliable statistical analyses. We were mostly interested in identifying factors that could help people remain stable compared to people who had indicated that their mental health had worsened compared to before the COVID-19 pandemic. The association between basic demographics and mental health change scores were analyzed using a Chi-square test for weighted data. A Chi-square test for weighted data was also used to analyze the association between basic demographics and positive outcomes, as well as background variables. Also, we analyzed whether social media use had changed, and a Chi-square test for weighted data was used to analyze the association between social media use and loneliness. Weighted logistical regression was used to test the association between a stable mental health and emotional loneliness, social loneliness, mental well-being and resilience, while controlling for gender, education level and age. Additionally, on measures of both emotional and social loneliness, we analyzed whether they differed between age groups using Tukey’s Honestly Significant Difference. For all analyses, an alpha of 0.01 was used for significance.

### 2.4. Ethical Considerations

The Central Committee on Research Involving Human Subjects in the Netherlands does not require approval from an ethical review committee for non-medical survey research, therefore this survey was exempt from ethical review. All respondents digitally signed an informed consent form before starting the survey.

## 3. Results

A total of 1519 participants completed the survey. There were no missing values. Participants’ ages ranged from 18 to 91 years (M = 53, SD = 16) and 52% of respondents were female (*n* = 789). Almost a third of participants were lived alone (29%, *n* = 438), and 23% (*n* = 351) had a lower educational level, 41% (*n* = 627) had a moderate level of education, and 36% (*n* = 541) had a higher level of education. Few respondents had contracted COVID-19 (2%, *n* = 37); however, an additional 4% of participants thought they had had COVID-19 (*n* = 59), but this was not confirmed by a test. Adverse events during the peak of the COVID-19 pandemic were experienced by 21% (*n* = 319) of respondents. The majority (*n* = 294) mentioned specific adverse events which were diverse. A relative or loved one passing away was mentioned by 40% of respondents (*n* = 118), of which 26% (*n* = 31) specifically indicated that the death was related to COVID-19. Other frequently mentioned adverse events included physical illness (other than COVID-19) that they themselves or a loved one experienced (19%, *n* = 57) and issues related to the protective measures of physical distancing (18%, *n* = 53), such as the inability to attend funerals, as well as more generally not being able to visit loved ones and friends for important social events. Lay-offs or loss of work assignments were mentioned in 8% (*n* = 23) of the cases. Social media use remained the same for most participants compared to usage before the COVID-19 pandemic (70%, *n* = 1059). An overview of descriptive statistics can be found in Table 1.

### 3.1. Changes in Mental Health

Approximately 80% of respondents rated their current mental health status the same (i.e., stable) as before the COVID-19 pandemic (*n* = 1207). Moreover, those people with a self-reported stable mental health mostly scored themselves as having a good or very good mental health (*n* = 1008, 84%). The Chi-square test showed that both male gender, χ^2^(1, *n* = 1427) = 9.09, *p* = 0.01, and no experience of adverse events, χ^2^(2, *n* = 1427) = 37.26, *p* < 0.001, were significantly associated with stable mental health status. The results of the Chi-square test are summarized in Table 2.

Logistic regression showed that social loneliness (*feeling a lack of relationships*: OR 1.22, 95% CI [1.08, 1.38], *p* = 0.001), emotional loneliness (*not feeling connected to loved ones*: OR 0.79, 95% CI [0.71, 0.87], *p* < 0.001) and well-being (OR 1.15, 95% CI [1.09, 1.22], *p* < 0.001) are significantly associated with having a stable mental health status. There was a significant moderate correlation between resilience and mental well-being (r = 0.56, *p* < 0.001). Logistic regression without including well-being showed that resilience was significant (OR 1.06, 95% CI [1.02, 1.11], *p* = 0.004).

Respondents in the age category 60–74 scored significantly lower on emotional loneliness than participants between 18 and 29 (*p* = 0.003) and 30 and 49 years old (*p* = 0.002), indicating that they experienced lower levels of emotional loneliness. On the subscale measuring social loneliness, results indicated that 18–29-year-olds scored significantly lower than participants who were 50–59 years old (*p* = 0.003), 60–74 years old (*p* = 0.003), and 75 years and older (*p* = 0.001). There was no significant relationship between social media use and emotional loneliness (*p* = 0.04) or social loneliness in the current group (*p* = 0.05).

### 3.2. Positive (and Negative Effects) of the COVID-19 Pandemic

More than half of the respondents (58%, n = 884) indicated that the COVID-19 pandemic also resulted in positive outcomes for them. The majority (92%, n = 814) mentioned specific positive outcomes, including: greater social connectedness (24%, n = 194) and rest (24%, n = 194). Additionally, working from home appeared to be a positive outcome for many participants (17%, n = 142).

For respondents who explicitly mentioned a positive outcome of the COVID-19 pandemic, only higher education was significantly associated with identification of positive outcomes from COVID-19, χ^2^(2, n = 1519) = 71.78, *p* < 0.001. The results are summarized in Table 3.

## 4. Discussion

In this study, we explored the mental health impacts of COVID-19 10 weeks after the start of the crisis in a cross-sectional survey, and 5 weeks after the relaxation of the measures. We were interested to see what part of participants reported worsened mental health at this point in the pandemic. Results were compared with the outcomes of similar surveys administered directly at the start of the crisis. Additionally, we aimed to learn about risk and protective factors for self-reported mental health during the pandemic, and explicitly asked participants whether they were able to name any positive effects of the pandemic.

### 4.1. Change in Mental Well-Being

Results showed that most people (80%) reported that their mental health was at the same level as their overall mental health status before the pandemic started, thus reporting a stable mental health. A prior Dutch study found that at the start of the pandemic, about one-third reported mental health deterioration, and thus our results suggest that mental health of participants improved as the pandemic persists [6].

Our results are in line with a longitudinal study in the UK [7], that found that over time, feelings of anxiety, entrapment and defeat decreased when compared to the start of the pandemic. Still, they found that depressive symptoms remained stable and suicide ideation actually increased over time. In our sample, we found a similar percentage of participants who reported suicide ideation when compared to the UK study (9.6% versus 9.8%). This rate is higher than the 2.8% normally reported in population-based surveys [15].

A self-reported stable mental health state was associated with being male and having a high level of well-being and inversely associated with emotional loneliness and adverse life events. This is in line with several other studies that indicated that loneliness and being female were risk factors for mental health problems during the pandemic [5,7].

An unexpected finding was that social loneliness was positively associated with having a stable mental health status according to self-report. It may be that physical distancing measures objectively reduced social contact, but as long as the meaningful connections are still felt, this may not have had an impact on loneliness or contributed to changes in mental health symptoms (e.g., depressive symptoms) for some participants. Emotional loneliness was highest among younger respondents (18–29) and then slightly decreased as age increased goes up, with the exception of the oldest age group (75+). Social loneliness, on the other hand, was lowest for the youngest age group and slightly increased with age. The younger participants in the sample might have undergone more substantive changes with emotional consequences during an important period of social development, including closure of schools and extracurricular activities, which may have contributed to an experience of more relevant changes in their lifestyle, as indicated by experiencing more emotional loneliness. Social loneliness may not have been experienced as much in this younger age category, which may reflect technological know-how and experience with social media and online interactions [16]. It may be that older people have less access to or familiarity with digital platforms to maintain social connections, which could lead to a greater sense of perceived social loneliness [17]. Results for the effect of education on self-reported mental health stability have been mixed during previous epidemics [18,19], while our study showed no effect of education.

Well-being was positively associated with self-reported mental health stability, but not resilience. This is in contrast with a previous study that found that both resilience and well-being were significantly associated with mental health during COVID-19 [20]. However, their study did not take into account mental health changes during the COVID-19 pandemic or after relaxation of measures. Our study did find a positive correlation between mental well-being and resilience indicating a close relationship between the two constructs. Previous longitudinal studies have shown that mental well-being positively affects resilience for mental health [21,22,23]. This suggests that well-being might be an important component or even precursor of resilience [24]. Due to the cross-sectional design of the current study, it was out of the scope of the study to detect how these constructs might interact to affect changes in mental health.

Although we see an improvement of self-reported mental health in our study, it is still unclear how people will react when restrictions are tightened again. It might be that people remain resilient, and mental health will remain similar as during our study. However, it might also be that people lose faith and become more desperate, resulting in a worsening of mental health. As we can expect restrictions to be tightened and relaxed in the coming months, continuous monitoring of mental health remains important to gauge both short and long-term effects.

### 4.2. Positive Effects

More than half of the participants mentioned positive outcomes of the COVID-19 pandemic. This is comparable to research on the SARS outbreak in 2003, in which most participants indicated that they could identify both negative and positive outcomes of the outbreak, even when they were directly affected by SARS [25]. Results indicated that seeing positive outcomes of the COVID-19 pandemic was associated with having a higher education, but not with gender or an adverse life event during the pandemic. Working from home was often mentioned by participants as a positive outcome. This might be due to reduced commute times (and fewer traffic jams), more perceived control and autonomy, greater efficiency [26] or more time with family and for loved ones. Research by Allen et al. [27] showed that certain people might prefer working from home due to the nature of their profession, such as jobs that require little interaction. On the other hand, work-related issues as a result of COVID-19 governmental measures were also mentioned as an adverse event by 8% of the participants. Despite physical distancing measures, most mentioned an increased sense of social connection and more meaningful connections as a positive outcome. This is in accordance with previous research indicating that, during the SARS pandemic, most people felt as though the support from family and friends was greater during the pandemic [28]. Additionally, a study among Dutch students also indicated a decrease in loneliness [8]. This could be another reason why so many people in our sample did not note a change in mental health status at the time of the survey compared to before the pandemic started. Telecommunications (via social media or digital applications) can help to maintain social connectedness despite physical distancing measures.

Surprisingly, social media use had not changed according to most people compared to before the COVID-19 pandemic, and changes in social media use were also not significantly associated with participants’ current level of emotional or social loneliness. This is not in line with other studies that reported an increase in social media use during the COVID-19 pandemic, and an association with loneliness, anxiety and depression [29,30,31]. It might be that this mainly holds for younger participants, as the age of participants in these studies was much lower. As the average age of our sample was 53, our sample population might be less-active users of social media, and therefore, their usage was not affected by the current crisis.

Overall, it is vital to understand how the pandemic and its secondary consequences impact the population, and what factors support resilience in the face of these adverse stressors. Even though most people experiencing an adverse life event did not necessarily face poorer self-reported mental health outcomes, it is important to be aware that some deterioration of mental health may occur. Certain groups might be particularly vulnerable for adverse outcomes, such as frontline workers exposed to COVID-19 [1], people who already have poorer mental health [1] or people who are survivors of COVID-19 [32]. This calls for having awareness and support options available in case of a new local outbreak, additional peaks of the virus, and the potential social and economic consequences of the pandemic, which might become more noticeable in the long term.

### 4.3. Limitations

The current study used a cross-sectional design in which antecedent data were collected retrospectively, and therefore causal inferences are not possible or might be subject to recall bias. Additionally, self-report measures were used of mental health. As such, people need to be able to reflect on how their mental health fluctuates [33] and this can be problematic. Peoples’ appraisal of emotional events are subjective to change of time, especially if measured with a delay [34].

The time period of the SWEMWBS was adapted from two weeks to four weeks, as was already mentioned in the Method section of this paper. This could have affected the level of validity to a certain extent, as all the validity studies that have been undertaken so far have been completed on the two-week response times [35].

Although poststratification weights were used, the survey may not have been totally representative of the adult Dutch population with regard to variables other than gender and age, and population segments that were affected by the novel corona outbreak may have been under- or over-represented in the survey.

Studies indicate a difference between the initial mental health response to COVID-19 and the lockdown restrictions. In this study, we did not differentiate between the two. Additionally, restrictions in the Netherlands had just been relaxed five weeks prior to the survey. As such, the current study cannot make any statements regarding how mental health might have varied since the start of the pandemic and the time period before the relaxations of measures. Given the cross-sectional design of the current study, we are not able to further disentangle the impact of COVID-19, the lockdown and the current relaxations. Future studies using longitudinal design are necessary to better understand the mechanisms of mental health impacted during a long-term crisis, like the COVID-19 crisis.

Lastly, our results do not generalize regarding people who are themselves survivors of COVID-19, as this group is underrepresented in our study (2%). Results from China and Italy indicate that these people show higher scores of depression, anxiety and PTSD [36,37]. Specific research into this group is needed.

## 5. Conclusions

This study shows that, ten weeks after the start of the crisis and five weeks after relaxation of restrictions, peoples’ self-reported mental health was better when compared to outcomes of surveys at the start of the crisis. Despite the unprecedented and uncertain circumstances caused by the novel coronavirus outbreak, people were able to perceive positive outcomes, and the majority reported stable and high levels of mental health and well-being.

## Figures and Tables

**Table 1 ijerph-17-09073-t001:** Demographic data and mean value total scores for mental health and effects of pandemic.

	Mean (SD)	Min	Max	*n* (%)
**Age**	53.10 (16.52)			
**Mental well-being**	25.44 (4.15)	7	35	
**Resilience**	20.62 (4.42)	6	30	
**Loneliness**				
Social loneliness	2.02 (1.85)	0	5	
Emotional loneliness	2.30 (2.02)	0	6	
**Suicide ideation**				144 (10%)
**Mental health**				
Stable				1207 (80%)
Decreased				92 (6%)
Increased				220 (15%)
**Social media use**				
Stable				1059 (70%)
Decreased				77 (5%)
Increased				383 (25%)
**Positive outcomes**				884 (58%)
**Adverse events**				319 (21%)

**Table 2 ijerph-17-09073-t002:** Association between change in mental health and background characteristics.

	Stable Mental Health Status	*p*-Value ^1^
**Gender**		0.01
Males	598 (87%)	
Females	609 (82%)	
**Adverse event**		<0.001
Yes	220 (73%)	
No	987 (88%)	
**Education**		N.S
Low	295 (88%)	
Middle	502 (85%)	
High	401 (82%)	

^1^ alpha set to 0.01. N.S = not significant. Percentages represent row percentages.

**Table 3 ijerph-17-09073-t003:** Association between positive outcome and background characteristics.

	Positive Outcome	*p*-Value ^1^
**Gender**		N.S
Males	412 (56%)	
Females	472 (60%)	
**Adverse events**		N.S
Yes	194 (60%)	
No	690 (58%)	
**Education**		<0.001
Low	164 (47%)	
Middle	337 (54%)	
High	383 (71%)	

^1^ alpha set to 0.01. N.S = not significant. Percentages represent row percentages.

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
