# Peer review of "The Bittersweet Effects of COVID-19 on Mental Health: Results of an Online Survey among a Sample of the Dutch Population Five Weeks after Relaxation of Lockdown Restrictions"

_ijerph, 2020, doi:10.3390/ijerph17239073_

Round 1
Reviewer 1 Report
The Authors answered to reviewers' comments in an appropriate manner. No further shortcomings to report.
Author Response
Summary of main editing:
We have completed an extensive spelling and grammar check. Additionally, we have made clear that mental health status was measured by self-report to ensure more accurate formulations throughout the manuscript.
The comments of reviewer 1 and 2 are shown in blue letters and our responses are in straight black letters with our adjusted text added in black cursive letters.
On behalf of the authors,
Mandy Gijzen.
Reviewer 1.
The Authors answered to reviewers' comments in an appropriate manner. No further shortcomings to report.
We would like to think reviewer 1 for giving us the opportunity to greatly improve our manuscript with their valuable comments and suggestions.
Reviewer 2.
From my point of view, I have to say now: quite well done.
There remain just some small grammatical issues, e.g. line #23: where -> were.
We would like to thank reviewer 2 for pointing out our grammatical issues, and have adjusted several sentences in light of this.
Lines:
#23:
and five weeks after restrictions were relaxed.
#47-48:
Furthermore, a group of international experts warned for the risk of the increase of suicidal behavior due to worsened mental health due to the crisis [1].
#53:
They did notice a small rise in suicidal thoughts over time.
#61-62:
still about one thirds of the Dutch population would report worsened mental health when compared to before the crisis
#243-245:
This is in line with several other studies that indicated that loneliness and being female were risk factors for mental health problems during the crisis [5,7].
Some formulation can be improved to be more accurate, when based on subjective self-reports, e.g. lines #154-5: "people that had a worsened mental health status"
We agree with reviewer 2 that certain statements and formulations need to be more accurate and cautious as they are pertaining to self-report. In this regard we have adjusted several sentences.
Lines:
#152-155:
The participants with self-reported improved mental health were not further analyzed as this group was too small to perform reliable statistical analyses. We were mostly interested identifying factors that could help people remain stable compared to people that had indicated their mental health had worsened compared to before the COVID-19 pandemic.
#191-192:
Moreover, those people with a self-reported stable mental health mostly scored themselves as having a good or very good mental health (n = 1008, 84%).
#209-210:
More than half of the respondents (58%, n = 884) indicated that the COVID-19 pandemic also resulted in positive outcomes for them.
#227-228:
Additionally, we aimed to learn about risk and protective factors for self-reported mental health during the crisis,
#242:
A self-reported stable mental health state was associated with being male and having a high level of well-being, and inversely associated with emotional loneliness and adverse life events.
#246-247:
An unexpected finding was that social loneliness was positively associated with having a stable mental health status according to self-report.
#261-262:
the effect of education on self-reported mental health stability have been mixed during previous epidemics [18,19], while our study showed no effect of education.
#263:
Wellbeing was positively associated with self-reported mental health stability, but not resilience.
#273-274:
Although we see an improvement of self-reported mental health in our study, it is still unclear how people will react when restrictions are tightened again.
#309-310:
Even though most people experiencing an adverse life event did not necessarily face poorer self-reported mental health outcomes, it is important to be aware that some deterioration of mental health may occur.

Reviewer 2 Report
From my point of view, I have to say now: quite well done.
There remain just some small grammatical issues, e.g. line #23: where -> were.
Some formulation can be improved to be more accurate, when based on subjective self-reports, e.g. lines #154-5: "people that had a worsened mental health status"
Author Response
Summary of main editing:
We have completed an extensive spelling and grammar check. Additionally, we have made clear that mental health status was measured by self-report to ensure more accurate formulations throughout the manuscript.
The comments of reviewer 1 and 2 are shown in blue letters and our responses are in straight black letters with our adjusted text added in black cursive letters.
On behalf of the authors,
Mandy Gijzen.
Reviewer 1.
The Authors answered to reviewers' comments in an appropriate manner. No further shortcomings to report.
We would like to think reviewer 1 for giving us the opportunity to greatly improve our manuscript with their valuable comments and suggestions.
Reviewer 2.
From my point of view, I have to say now: quite well done.
There remain just some small grammatical issues, e.g. line #23: where -> were.
We would like to thank reviewer 2 for pointing out our grammatical issues, and have adjusted several sentences in light of this.
Lines:
#23:
and five weeks after restrictions were relaxed.
#47-48:
Furthermore, a group of international experts warned for the risk of the increase of suicidal behavior due to worsened mental health due to the crisis [1].
#53:
They did notice a small rise in suicidal thoughts over time.
#61-62:
still about one thirds of the Dutch population would report worsened mental health when compared to before the crisis
#243-245:
This is in line with several other studies that indicated that loneliness and being female were risk factors for mental health problems during the crisis [5,7].
Some formulation can be improved to be more accurate, when based on subjective self-reports, e.g. lines #154-5: "people that had a worsened mental health status"
We agree with reviewer 2 that certain statements and formulations need to be more accurate and cautious as they are pertaining to self-report. In this regard we have adjusted several sentences.
Lines:
#152-155:
The participants with self-reported improved mental health were not further analyzed as this group was too small to perform reliable statistical analyses. We were mostly interested identifying factors that could help people remain stable compared to people that had indicated their mental health had worsened compared to before the COVID-19 pandemic.
#191-192:
Moreover, those people with a self-reported stable mental health mostly scored themselves as having a good or very good mental health (n = 1008, 84%).
#209-210:
More than half of the respondents (58%, n = 884) indicated that the COVID-19 pandemic also resulted in positive outcomes for them.
#227-228:
Additionally, we aimed to learn about risk and protective factors for self-reported mental health during the crisis,
#242:
A self-reported stable mental health state was associated with being male and having a high level of well-being, and inversely associated with emotional loneliness and adverse life events.
#246-247:
An unexpected finding was that social loneliness was positively associated with having a stable mental health status according to self-report.
#261-262:
the effect of education on self-reported mental health stability have been mixed during previous epidemics [18,19], while our study showed no effect of education.
#263:
Wellbeing was positively associated with self-reported mental health stability, but not resilience.
#273-274:
Although we see an improvement of self-reported mental health in our study, it is still unclear how people will react when restrictions are tightened again.
#309-310:
Even though most people experiencing an adverse life event did not necessarily face poorer self-reported mental health outcomes, it is important to be aware that some deterioration of mental health may occur.

This manuscript is a resubmission of an earlier submission. The following is a list of the peer review reports and author responses from that submission.
Round 1
Reviewer 1 Report
The most important critical point regarding this paper concerns the time-delay of the data mining. The authors reported (line 67) that they collected the data during the fifth week, i.e. four weeks after the Dutch government relaxed the measures.
Even though the authors mentioned this fact in the foremost part of Limitations (lines 247-8), my serious objection includes the question: what can be measured four to five weeks after the situation changed, when the authors used the measurement tools, which were developed for capturing of the present moment (11-item Loneliness Scale) or the participants´ experiences of each over the last two weeks (Warwick-Edinburgh Mental Wellbeing Scale)? Moreover, the authors claimed (line 105), that this scale “focuses on the positive of psychological functioning for the past four weeks” – this (1) contradicts the abovementioned statement (about last two weeks) of creators of this scale, and (2) even past four weeks would be not enough, because the governmental measures were relaxed just four weeks before the data-mining process started.
On one side, we were exhaustively informed, how were the participants asked to recall positive and negative outcomes of Covid-19 pandemic (part 2.2.2, lines 116-21).
On the other side, the key question of this study, the (subjective) rating of participants’ “mental health before the COVID-19 crisis and at this moment (i.e. during the COVID-19 crisis)” as noted on lines 88-9, arouses serious doubts about the correctness of the question, if asked as “at this moment (i.e. during the COVID-19 crisis)”, during the fifth week, i.e. four weeks after the governmental measures started to be subsequently relaxed.
Furthermore, some critical comments should be made on the Discussion section (e.g. absent references, weak explanations). However, if there are persistent doubts about the measurements (see above), it would be purposeless.
Reviewer 2 Report
This is an interesting paper on an important topic. I have no substantive criticisms. Generally it is very well written, but there are one or two colloquial phrases and at least one error. The latter is the phrase "...violate psychological responses" (line 255). The word "less" was used in a few places where "fewer" is more correct (e.g. lines 215 and 228). The phrase "people are okay with seeing less people" (line 215 and abstract) seems a bit colloquial for a scientific paper.
Reviewer 3 Report
The manuscript is on an interesting issue. The paper is well written, and certainly falls within the scope of the Journal. Minor shortcomings to report:
1) The age range of the sample is wide (18-91 years) but the Authors stated that a weighted logistical regression was used to test the association between a stable mental health and several variables, controlling for gender, education level and age. However, I was wondering if they found any qualitative difference in terms of social/emotional loneliness in the younger subjects of the sample who underwent to the more relevant changes in their lifestyle (see the lockdown of schools, cultural institutions, cinemas, etc etc).
2) I would like to know if information was collected on the use of social media and more in general on the availability of Internet connections as factors involved in the levels of perceived social or emotional loneliness. Apparently, no importance has been given to this aspect, except maybe for smart working.
3) In the conclusions' section the Authors stated that people were with a stable mental health despite health and economic shocks. How economic shock was rated or quantified ? Was this aspect quantified at all ?
Round 2
Reviewer 1 Report
The authors have made a lot of work in order to improve the manuscript, according to the comments of the reviewers. However, there still occur weak points in the current version:
- Line 4: please, specify “shortly” more properly. Actually, “shortly” seems to be a bit undefined (throughout the entire manuscript)…
- Lines 57-8: I appreciate this new addition, but as we can see later (e.g. Table 1), the majority (70%) of respondents reported no change in using of social media, what contrasts with some other resources from the given field (however, no such other data are cited or discussed in the manuscript)
- Lines 63-85: this is significantly re-worked part, but, unfortunately, it contains some serious weak points:
-> (a) The authors mentioned, that “prior studies on reactions to adverse life events have shown that mental health issues arising from experiences with the event may only be temporary in duration for the majority of the population [14, 15]”
-> (b) Furthermore, the authors quoted, that “that initial psychosocial problems during the early stages of COVID-19, rapidly decreased after three days [16].
-> (c) Another two cited resources claimed that “that web-searches for mental health spiked right at the start of the pandemic, but quickly stabilized after government action to introduce measures to flatten the curve of the virus [17] and „after the initial shock due to COVID-19, people adjusted over time to the lockdown restrictions [18].
Despite this argumentation (a, b, c), the authors expected that they can identify “short and long-term impacts of the pandemic on mental health“ when asking the general population during the fifth week after the measures were relaxed. Unfortunately, the manuscript does not present any typology of short and long-term impacts of the pandemic on mental health.
How can I understand to the sentence (lines 78-9): “However, despite these relaxations, the COVID-19 crisis was still very much present in the Netherlands.“, when the measures were relaxed? (moreover, the request to work from home was considered as advantage by many – compare with lines 218-9).
Why did you “expected that the crisis still have a profound impact on mental wellbeing” (lines 81-2), especially, when you cited previous studies (see points a, b, c above)? In this regard, there is nothing unexpectable on the main finding of this manuscript, i.e. that the „Most participants (80%, n = 1207) reported no change in mental health since the COVID-19 pandemic“ (lines 26-7).
- Lines 128-9: although the authors added the information that “The time period of the questionnaire was changed from two weeks to four weeks“, the methodological consequences of this act were not discussed.
- Line 189: I guess they should be excluded, if not related to Covid-19 (?)
- Lines 237-43: The authors claimed “that most people reported that their mental health had not changed during the COVID-19 pandemic compared to their general mental health before the pandemic“ what is accompanied with the text „Our results are in line with recent findings from surveys carried out in England , showing that although the COVID-19 pandemic itself had a negative impact on mental health, the lockdown measures were mostly associated with improvement in subjective well-being.“ I don’t think so, because the authors of this paper did not declared the worsening of mental health, i.e. the authors did not find that the metal health was worsened during the lockdown, they just anticipated it!
- Lines 310-2: “Future studies would need to look at how social connections through telecommunications, such social media, affects feelings of loneliness taking into account age groups as well.” – there exist many of such studies…
- The Conclusions seems to be too vague, but, frankly speaking, maybe there is not much to be mentioned: the authors assumed that the mental health was affected during the lockdown, they „measured“ it retrospectively during the fifth week after measures relaxed, utilizing the self-report instrument (designed for the time period of past two weeks) and they found that 80% reported no changes in their mental health…
- Results showed that most people reported that their mental health had not changed during the COVID-19 pandemic compared to their general mental health before the pandemic – but, what if it has changed, but the respondents are (yet) not capable to reflect it?